# Effects of Cellulase and Lactic Acid Bacteria on Ensiling Performance and Bacterial Community of *Caragana korshinskii* Silage

**DOI:** 10.3390/microorganisms11020337

**Published:** 2023-01-29

**Authors:** Baochao Bai, Rui Qiu, Zhijun Wang, Yichao Liu, Jian Bao, Lin Sun, Tingyu Liu, Gentu Ge, Yushan Jia

**Affiliations:** 1Key Laboratory of Forage Cultivation, Processing and High Efficient Utilization of Ministry of Agriculture and Rural Affairs, Inner Mongolia Agricultural University, Hohhot 010019, China; 2Key Laboratory of Grassland Resources, Ministry of Education, Inner Mongolia Agricultural University, Hohhot 010019, China; 3Inner Mongolia Academy of Agricultural & Animal Husbandry Sciences, Hohhot 010031, China; 4College of Agriculture, Inner Mongolia Minzu University, Tongliao 028000, China

**Keywords:** lactic acid bacteria, cellulase, microbial community, fermentation characteristics, *Caragana korshinskii*

## Abstract

The aim of this study was to evaluate the effects of cellulase (CE) and lactic acid bacteria (LAB) on *Caragana korshinskii* silage by analyzing the fermentation parameters, chemical composition, and bacterial community. The *Caragana korshinskii* was harvested at the fruiting period and treated with cellulase and LAB alone as a control treatment with no additive (CK). The ensiling performance and bacterial community were determined after 3, 7, 15, 30, and 60 days of fermentation process. Compared with the CK group, the pH, dry matter loss, and ammonia nitrogen content were significantly (*p* < 0.05) decreased in the LAB and CE treatments. Compared with the CK and LAB group, the contents of acid detergent fiber, neutral detergent fiber, and acid detergent lignin in the CE group decreased significantly (*p* < 0.05), and the water-soluble carbohydrates, acetic acid, and lactic acid concentrations increased significantly (*p* < 0.05). At the genus level of microorganisms, the addition of cellulase and LAB significantly reduced the microbial diversity. Compared with the CK group (78.05%), the relative abundance of *Lactiplantibacillus* in the CE group (90.19%) and LAB group (88.40%) significantly (*p* < 0.05) increased. The relative abundance of *Pediococcus* in the CE group (3.66%) and LAB group (2.14%) was significantly (*p* < 0.05) lower than that in the CK group (14.73%). Predicted functional profiling of 16S rRNA genes revealed that the addition of cellulase and LAB increased the pyruvate metabolic pathway during *Caragana korshinskii* silage, thereby increasing the accumulation of lactic acid concentration. The addition of cellulase expressed a better advantage in the biosynthetic capacity of lysine. In summary, the addition of cellulase and LAB could adjust the bacterial community to improve the silage quality of *Caragana korshinskii*, and the addition of cellulase exhibited better results than the LAB additives.

## 1. Introduction

The Xing’an League region of the Inner Mongolia Autonomous Region is located in the hinterland of Horqin sandy land. Natural meadows are rare, and the main source of local beef crude feed is corn stover. The shortage of quality forage grasses has severely constrained local livestock development. Therefore, there is an urgent need to develop more available feed resources as animal feed. Recently, *Caragana korshinskii* has been widely cultivated locally as a renewable resource with the implementation of ecological projects such as grazing prohibition and sand control [1]. *Caragana korshinskii* is a genus of *Caragana* in the legume family. The genus *Caragana* is a dominant species in cold-temperate arid scrub, desert, and mountain meadows, and is well known in China for its role in sand fixation, soil and water conservation, nectar, fuel, and fodder. To date, more than 100 species of *Caragana* have been found worldwide, mainly in arid and semi-arid regions of Asia and Europe, while at least 60 species of *Caragana* have been found in China alone. Among these plants, *Caragana korshinskii* is the most common plant in the arid and semi-arid regions of the Inner Mongolia Plateau [2]. In China, more than 4 million tons of *Caragana korshinskii* are harvested annually to alleviate plant aging [3]. *Caragana korshinskii* has a high protein content and is rich in several amino acids, especially lysine, which is an essential amino acid in beef cattle, and therefore has a high value for feeding [4]. However, the harvested *Caragana korshinskii* was treated as fuel, causing a huge feed waste.

Fresh *Caragana korshinskii* containing tannins and stipule thorns that affect palatability is not eaten by beef cattle. Silage, as a microbial anaerobic fermentation technique, can soften spines and produce an aromatic odor, which can improve the palatability of *Caragana korshinskii*. Silage technology mainly refers to the proliferation of lactic acid bacteria under anaerobic conditions, using soluble carbohydrates to produce large amounts of lactic and acetic acids, thereby reducing pH, inhibiting the growth of undesirable microorganisms, preserving feed, and improving palatability [5]. Considering the high lignocellulose content and low soluble carbohydrate content in fresh *Caragana korshinskii*, cellulase could be added. Cellulase increases the content of water-soluble carbohydrates (WSC) by degrading cellulose and provides a substrate for lactic acid production by lactic acid bacteria [6]. Furthermore, He et al. [7] found that the addition of cellulase increased the true protein content, decreased the fiber and ammonia nitrogen (NH_3_-N) contents, and improved the silage quality of mulberry leaf silage. In addition to increasing the soluble carbohydrate content of fresh *Caragana korshinskii*, the number of epiphytic lactic acid bacteria can be increased to obtain successful silage. Lactic acid bacteria, as one of the most common additives, is used to obtain successful silage products. Lactic acid bacteria can enhance homologous lactic acid fermentation, produce a large amount of lactic acid, and rapidly reduce the pH value, thus inhibiting adverse microorganisms and obtaining better fermentation quality [8]. Ogunade et al. [9] found that in alfalfa silage production, some harmful microorganisms decrease feed quality and lactic acid bacteria improve silage quality by competing with these microorganisms to rapidly eliminate pathogens.

To date, the effects of cellulase and LAB as inoculants on the silage fermentation characteristics, chemical composition, complex microbial communities, and microbial interactions of various types of grass have been reported. However, the effects of cellulase and LAB on the silage fermentation of *Caragana korshinskii* remain unclear. Therefore, this study aimed to evaluate the effect of cellulase and LAB additives on *Caragana korshinskii* silage by analyzing the fermentation parameters, chemical composition, and bacterial community.

## 2. Materials and Methods

### 2.1. Silage Preparation

*Caragana korshinskii* was harvested during the fruiting period in Bayannur Sumu, Horqin Right Wing Central Banner, Xing’an League, Inner Mongolia Autonomous Region (44°77′ E, 121°86′ N). The raw material was harvested by Caragana Harvester, cut to 2–3 cm, and then processed into silky objects by a korshinskii kneading machine. The moisture content of the raw material was 46.3%. After weighing 16.76 kg of raw materials, 5.74 kg of distilled water was added and stirred well to control the moisture content at 60%. The uniformly stirred *Caragana korshinskii* was divided into three groups, and treated with the following: (1) no additive control (CK); (2) cellulase (CE; Yidu Biological Technology Co., Ltd., Hohhot, Inner Mongolia, China; mainly including exo-β-glucanase, endo-β-glucanase, β-glucosidase, and xylanase; CE activity as 5 × 10^5^ U/g; addition amount: 1 × 10^5^ U/kg fresh material (FM)); (3) lactic acid bacteria (LAB; Yidu Biological Technology Co., Ltd., Hohhot, Inner Mongolia, China; LAB application rate: 1 × 10^6^ cfu/g FM). All the additives were mixed homogeneously with *Caragana korshinskii*. Each treatment batch was divided into three copies, each 500 g, packed in polyethylene bags (30 × 40 cm), and vacuum sealed. A total of 45 bags were prepared (3 treatments × 3 repetitions × 5 silage days) and stored at ambient temperature (20–30 °C). After 3, 7, 15, 30, and 60 days of ensiling, the chemical composition, fermentation characteristics, and bacterial community were analyzed.

### 2.2. Analysis of Fermentation Parameters, Chemical Composition, and Microbial Counts

In the fresh *Caragana korshinskii* and silage, 10 g of forage samples were blended with 90 mL of sterilized water, and organic acid liquid was obtained by beating the homogenate for 2 min. The pH was determined by an electrode pH meter (PHS-3C, INESA Scientific Instrument Co., Ltd., Shanghai, China). The concentrations of organic acids, lactic acid (LA), acetic acid (AA), propionic acid, and butyric acid were measured using high-performance liquid chromatography according to the method of Cheng et al. [10]. The ammonia nitrogen content was measured by phenol-sodium hypochlorite colorimetry determination [11]. Lactic acid bacteria were cultured and counted on an MRS medium. Aerobic bacteria were cultured and counted using a nutrient agar medium. Molds and yeasts were cultured and counted on potato dextrose agar medium. Coliform bacteria were cultured and counted on eosin methylene blue medium. All four culture media were purchased from Guangdong Huankai Microbial Sci. and Tech. CO., Ltd (Guangzhou, China). Colonies were calculated as the viable number of microorganisms and transformed into logarithmic scales based on FM basis (Log_10_ cfu/g FM).

Each group of samples was tested 3 times in parallel for chemical analysis. Fresh or ensiling materials were dried in a fan-forced oven at 65 °C for 48 h to analyze DM content [12]. The crude protein (CP) content was determined by Kjeldahl method [13]. The WSC content was measured using the anthrone colorimetry method [14]. The contents of acid detergent lignin (ADL), acid detergent fiber (ADF), and neutral detergent fiber (NDF) were measured by an A220 fiber analyzer according to the method of Van Soest et al. [15].

### 2.3. DNA Extraction, PCR and Sequencing

Fresh sample and silage samples were analyzed by Biotechnology platform (LC-Bio Co., Ltd., Zhejiang, China). The cetyltriethyl ammonium bromide (CTAB) method was used to extract the total DNA of microbial groups from various sources [16], and the quality of DNA extraction was detected by agarose gel electrophoresis. At the same time, DNA concentration was quantified by UV–vis spectrophotometer. The hypervariable region V3–V4 of the bacteria 16S rRNA gene were amplified with primers 341F(5′-CCTACGGGNGGCWGCAG-3′) and 805R(5′-GACTACHVGGGTATCTAATCC-3′). The PCR products were purified using the AMPure XT beads (Beckman Coulter Genomics, Danvers, MA, USA), quantified using Qubit (Invitrogen, Carlsbad, CA, USA), then mixed according to the sequencing requirements of each sample. PCR products were detected by 2% agarose gel and recovered by AMPure XT beads recovery kit. The PCR products were sequenced on Agilent 2100 biological analyzer (Agilent Technologies, Inc., Santa Clara, CA, USA) and an Illumina platform (Kapa Biosciences, Woburn, MA, USA) and after edulcoration and ration.

### 2.4. Bacterial Community Sequencing Analysis

The raw reads were carried out quality filtering by the software of fqtrim (v0.94) (http://ccb.jhu.edu/software/fqtrim/ (accessed on 1 September 2022)) under specific filtering conditions so that gain high quality clean labels. Using Vsearch software (v2.3.4) (https://github.com/torognes/vsearch (accessed on 1 September 2022)) to filter chimeric sequences. Alpha diversity and beta diversity were reckoned by random normalization to the same sequence using QIIME2 [17]. Microbial relative abundance was used to represent bacterial classification. Based on 16S rRNA data, PICRUSt was used to predict metabolic genes [18]. The Kyoto Encyclopedia of Genes and Genomes (KEGG) was also used to assign genes to metabolic pathways. High-throughput sequencing data were analyzed and most graphs were plotted using the https://www.omicstudio.cn/tool (accessed on 15 September 2022) Omic Studio tool. The Bar plots was made in GraphPad prism 8.0.2.

### 2.5. Statistical Analysis

The fermentation parameters, chemical composition, and microbial community were calculated by two-way analysis of variance. The statistical model is as follows:Y_ıjh_ = *μ* + *α_ι_* + *β_j_* + *αβ_ıj_* + *ϵ_ıjh_*(1)where Y_ıjh_ is an observation, *μ* is the overall mean, *α_ι_* is the effect of additives (*ι* = CK, CE, LAB), *β_j_* is the number of ensiling days (*j* = 3, 7, 15, 30, 60), *αβ**_ıj_* is the additives × number of ensiling days interaction, and *ϵ_ıjh_* is the error. The Duncan’s tests were used to separate significant differences and was considered statistically significant when the probability was less than the 5% level.

## 3. Results

### 3.1. Chemical Composition and Microbial Counts of Fresh Caragana korshinskii

The chemical compositions and microbial counts of *Caragana korshinskii* before ensiling are presented in Table 1. The DM content of fresh *Caragana korshinskii* was 463.80 g/kg of FM and the contents of NDF, ADF, ADL, CP, and WSC were 706.92, 594.99, 155.71, 102.53, and 27.69 g/kg DM, respectively. Microbial counts in the *Caragana korshinskii* for lactic acid bacteria, yeasts, coliform bacteria, aerobic bacteria, and molds were 6.11, 8.61, 7.35, 8.15, and 2.82 Log_10_ cfu/g FM.

### 3.2. Fermentation Characteristics and Chemical Compositions of Caragana korshinskii Silage

The dynamic changes of fermentation characteristics and chemical compositions of *Caragana korshinskii* silages are displayed in Table 2 and Table 3, respectively. At each period of silage, the pH values of CE and LAB groups were significantly lower than that of CK group (*p* < 0.05) (Table 2). With the increase of silage days, the pH values of CE and LAB groups showed a decreasing trend, the pH values of CK group reached the lowest value at 15 days of ensiling, and the pH values increased slightly at 30 and 60 days. The concentrations of LA and AA in CE group was significantly higher than that in CK and LAB group at 60 days of ensiling (*p* < 0.05), and there was no significant difference in LA and AA concentrations among the three groups at the ensiling stages of 3 to 30 d. In addition, no propionic acid or butyric acid was detected in the three treatment groups throughout the silage process. The lowest NH_3_-N contents was observed in CE groups, and the NH_3_-N was not detected in CE group after 7 days of ensiling.

After 60 days of ensiling, the DM content in CE and LAB groups was significantly (*p* < 0.05) higher than that in CK group (Table 3). The WSC, NDF, ADF, ADL, and DM were significantly decreased (*p* < 0.05) in all treatment groups as the days of silage increased. The ADF, NDF, and ADL contents of CE group were significantly lower than CK and LAB groups at each period of silage (*p* < 0.05), while the WSC content of CE group was significantly higher than CK and LAB group at each period of silage (*p* < 0.05). At 60 days of ensiling, the content of CP in three groups was significantly (*p* < 0.05) higher than that before ensiling, and the content of CP in CE and LAB groups was significantly (*p* < 0.05) higher than that in CK groups.

### 3.3. Bacterial Diversity and Community Composition in Caragana korshinskii Silage

The alpha diversity of dynamic *Caragana korshinskii* silage bacterial communities is shown in Table 4. The bacterial coverage of each group was 0.99, indicating that the bacterial community analysis conducted at this sequencing depth was reliable enough. Ensiling days had a significant effect on OTUs, Chao1, Shannon, Simpson, and Pielou-e (*p* < 0.05), The Pielou-e, Simpson, Shannon, Chao1, and OTUs decreased with increasing ensiling days in all groups. There was no significant difference in Chao1 and OTUs indexes among the three treatment groups at days 60 of ensiling. The Shannon, Simpson, and Pielou-e indexes were significantly lower in the CE and LAB-treated silages than in the CK silage at days 60 of ensiling (*p* < 0.05). The changes in the microbial community of the silage were further verified by beta diversity analysis using principal coordinates analysis (PCoA) (Figure 1). Principal coordinates 1 (PCo1) and 2 (PCo2) explain 69.79% and 19.82% of the total variance in *Caragana korshinskii* silage, respectively. The bacterial communities of fresh and silage samples were significantly separated, while there was no significant difference between silage samples treated with additives, and three groups of silage samples gathered together.

The changes in bacterial communities at the phylum and genus levels for each treatment group are shown in Figure 2. In fresh samples, Proteobacteria (73.68%) and Cyanobacteria (16.31%) accounted for most of the advantages of the total phylum sequence (Figure 2A). The main genera attached to fresh *Caragana korshinskii* were *Erwinia* (25.32%) and Cyanobacteria_unclassified (16.31%), followed by *Pantoea* (14.59%) and *Escherichia* (10.10%) (Figure 2C). After ensiling, Firmicutes replaced Proteobacteria and Cyanobacteria as the dominant phylum. The bacterial community composition was similar in CE and CK groups, and the relative abundances of Firmicutes increased with the prolongation of silage time. The Firmicutes in the LAB group decreased slightly after 7 days of ensiling and the Cyanobacteria increased slightly, but the relative abundance of the Firmicutes in each treatment group reached more than 90% at 60 days of ensiling. At the genus level of microorganisms, the main bacterial community composition of CK and CE groups was different from that of LAB group. During the first 15 days of ensiling, the CK and CE groups were dominated by *Pediococcus* and after 30 days of ensiling by *Lactiplantibacillus*. *Lactiplantibacillus* had always occupied a dominant position in the LAB group during the whole silage process. The abundance of *Lactiplantibacillus* was higher in the CE (90.19%) and LAB groups (88.40%) at 60 days of ensiling compared to the control group (78.05%). Figure 2B shows the one-way ANOVA of phylum level between FM and silage groups after 60 days of *Caragana korshinskii* ensiling, and there was a significant difference between FM and silage in the phylum Firmicutes and Proteobacteria (*p* < 0.05). One-way ANOVA of genus level showed that *Lactiplantibacillus*, *Pediococcus*, *Enterobacter,* and *Klebsiella* had significant (*p* < 0.05) differences between FM and silages (Figure 2D). The additive treatment groups effectively reduced the abundance of *Pediococcus* and increased the abundance of *Lactiplantibacillus* compared to the control group.

As shown in Figure 3, the linear discriminant analysis effect size (LEfSe) was applied to evaluate the difference in microbial community among the three groups at different silage times (LDA score > 3.0). We focused on the differences among the three groups at the genus level. After 3 days of ensiling, the *Lactiplantibacillus* was higher in the LAB group, while *Bacillus* and *Serratia* were higher in CE group, and *Arsenophonus* was higher in CK group. After 7 days of ensiling, *Sphingomonas* was higher in CE group. After 15, 30 and 60 days of silage, the main differential microorganisms were only present in the CK and LAB groups. *Curtobacterium*, *Eggerthella*, *Catonella*, *Weissella*, and *Pediococcus* were higher in CK group. *Nevskia* and *Flavonifractor* were higher in LAB group.

### 3.4. Relationships between Fermentation Parameters and Bacterial Community

The correlation between the fermentation quality of *Caragana korshinskii* silage and bacterial community was further investigated using redundancy analysis (RDA). The relationship between bacterial kinetics and silage quality for each genus is shown in Figure 4. In the fermentation quality, the genera *Lactiplantibacillus* was positively correlated with LA and AA content and negatively correlated with pH. The genera *Escherichia*, *Enterobacter*, and *Klebsiella* were positively correlated with NH_3_-N content. In the chemical compositions, the genus *Lactiplantibacillus* was negatively correlated with DM, NDF, ADF, ADL, and WSC content, and the genera *Enterobacter* and *Escherichia* were negatively correlated with CP content.

### 3.5. Predicted Metabolic Pathways on Three Levels

The metabolic function profiles by 16S rRNA gene-predicted for fresh and silages at 60 days is shown in Figure 5. On the first pathway level (Figure 5A), the relative abundance of metabolism was apparently higher than other pathways, and metabolism was promoted after ensiling. On the second pathway level (Figure 5B), the main metabolism pathways were carbohydrate metabolism, amino acid metabolism, nucleotide metabolism, energy metabolism, and metabolism of cofactors and vitamins. The carbohydrate metabolism, amino acid metabolism, nucleotide metabolism, and metabolism of cofactors and vitamins were promoted after ensiling, whereas the energy metabolism was inhibited after ensiling. As seen in Figure 5C, the glycolysis/gluconeogenesis and pyruvate metabolism were the two most dominant carbohydrate metabolism pathways, which were promoted after ensiling. The pyruvate metabolism in CE and LAB group was higher than that in CK group. Most amino acid metabolism was inhibited after ensiling, whereas lysine biosynthesis was significantly promoted after silage, with the highest lysine biosynthesis in the CE group (Figure 5D). As illustrated in Figure 5E, most energy metabolic pathways remained stable during ensiling, while the oxidative phosphorylation, photosynthesis, photosynthesis proteins, and photosynthesis–antenna proteins were reduced after ensiling. The porphyrin and chlorophyll metabolism changed the most in the metabolism of cofactors and vitamins pathway, which was promoted after ensiling (Figure 5F). The porphyrin and chlorophyll metabolism were significantly (*p* < 0.05) increased in the LAB group compared to the CK and CE groups.

## 4. Discussion

The pH value, organic acid, and ammonia nitrogen content are commonly used indicators to evaluate fermentation quality. In this study, we observed that the pH value of the CE and LAB groups was lower than those of the CK group and the LA concentration was higher than that of the CK group in all periods of silage, which indicated that the addition of cellulase and LAB could promote silage fermentation. Cellulase can degrade cellulose to WSC early in silage, providing substrate for lactic acid bacteria fermentation, resulting in rapid accumulation of LA and lower pH [19]. The addition of LAB accelerates the conversion of WSC to lactic acid and reduces pH in the early stages of silage [20]. It is noteworthy that the CK group showed an increase in pH at 30 and 60 days of ensiling, which we believe is related to the increase in *Escherichia* at 30 days of ensiling. *Escherichia* consumed protein to produce NH_3_-N, which neutralized the acid. The concentrations of LA and AA in the CE group were significantly higher than those in the CK and LAB groups after 60 days of ensiling. This may be due to the fact that the addition of cellulase can provide more substrates for lactic acid bacteria fermentation by degrading lignocellulose and increase the concentration of LA and AA [21,22]. The results of Kung et al. [23] indicated that the action of plant protein hydrolases may be the main reason for NH_3_-N production. The content of NH_3_-N in all three treatment groups were rarely likely because of a rapid decline in pH inhibited proteolytic activity [24]. The lowest NH_3_-N content was obtained in the CE group. Li et al. [25] also found that the addition of cellulase to cassava leaf silage can significantly reduce NH_3_-N production. All groups of NH_3_-N content were below the detection level at 60 days of ensiling, which may be due to some lactic acid bacteria can induce nitrification and convert ammonia nitrogen into nitrate nitrogen [26].

Good quality fermented silage is usually well preserved in terms of nutrients. After 60 days of ensiling, the CK group had the highest dry matter loss, which was because the CK group had the lowest WSC content, and when soluble sugar was lacking during fermentation, some lactic acid bacteria would shift from homofermentative to heterofermentative [27], and the DM consumption would increase when heterogeneous fermentation occurred in silage [28]. The cellulase and LAB additives effectively preserve the DM content. The CP content of CE group and LAB group was higher than that of CK group, which may be due to the lower pH value of these two groups than that of CK group, which helps to protect the protein from catabolism [29]. All three treatment groups showed slightly higher CP content after silage treatment than fresh samples, which may be related to the decrease in NDF content. Ebrahimi et al. [30] found an increase in CP content after oil palm leaf silage, which they interpreted as an increase in the relative CP content due to the loss of NDF. The high content of ADF and NDF in silage is not conducive to absorption and digestion of animals. The degradation of NDF and ADF was more pronounced in the CE group, and a study by Zhang et al. [31] also found that cellulase significantly reduced the levels of NDF and ADF. Throughout the silage process, CE group had the highest WSC content and the lowest NDF and ADF content, which were attributed to the ability of cellulase to degrade plant cell walls and produce more soluble sugars [32]. In terms of nutrient composition and fermentation parameters, the addition of both cellulase and LAB improved the quality of *Caragana korshinskii* silage, with cellulase being more effective.

In order to understand the differences of bacterial communities among the three treatments, we used 16S rRNA high-throughput sequencing technology. A total of 4,025,224 reads were obtained by high-throughput sequencing, ranging from 80,350 to 86,972 per treatment. The coverage of all samples is around 0.99, making it feasible to analyze the bacterial community. The alpha diversity can explain species richness, evenness, and diversity in bacterial communities. In the present study, the OUTs, Chao1, Simpson, Shannon, and Pielou-e indexes of all treatment groups decreased with the increase of silage time. This is because with the decrease of silage pH, the complex bacterial community in silage is gradually replaced by lactic acid bacteria [33]. At 60 days of ensiling, the CE and LAB groups significantly reduced the Simpson index, Shannon index, and Pielou-e index compared with CK group. Xiong et al. [34] found that the addition of cellulase and LAB could significantly reduce the Pielou-e, Shannon, and Simpson index of hybrid *Pennisetum* silage. This is because cellulase provides more soluble sugars for lactic acid bacteria and promotes mass reproduction of lactic acid bacteria, thereby increasing the proportion of lactic acid bacteria in silage microbial populations. Lactic acid bacteria produce large amounts of lactic acid, which lowers the pH of the feed and inhibits the proliferation of unfavorable microorganisms, thereby reducing bacterial diversity [35]. The beta diversity can reflect the difference of bacterial community in each treatment group or each individual. The PCoA diagram showed a clear separation of bacterial communities between fresh and silage samples, indicating that silage reconfigured the bacterial community. Similar results were reported by Zeng et al. [36] and Li et al. [37], who also found significant differences in bacterial communities in FM and silage.

The quality of silage is closely related to bacterial community composition [38]. In this study, we found that Proteobacteria and Cyanobacteria were the most abundant microorganisms in FM, the dominant position of Proteobacteria and Cyanobacteria in each treatment group was replaced by Firmicutes after ensiling. Mu et al. [29] reported similar results. The reason for this phenomenon may be that the main microorganisms involved in lactic acid fermentation belong to Firmicutes. Microorganisms of the Proteobacteria and Cyanobacteria are both Gram-negative bacteria. The permeability of the outer membrane of Gram-negative bacteria is usually affected by the pH value. The lactic acid produced by lactic acid bacteria reduces the pH value and thus inhibits the growth of Proteobacteria and Cyanobacteria [39]. It is worth noting that the relative abundance of Cyanobacteria in LAB group increased slightly after 30 days of ensiling, which is difficult to explain when the pH is less than 4.00. Feng et al.’s [40] research believed that the cause of this phenomenon may be that the detected Cyanobacteria are part of plant samples. The main epiphytic genera in fresh *Caragana korshinskii* were *Erwinia*, *Cyanobacteria*, *Pantoea*, and *Escherichia*. *Erwinia*, *Pantoea*, and *Escherichia* have parthenogenic anaerobic properties and belong to *Enterobacteriaceae*, they can compete with lactic acid bacteria for nutrients during silage, resulting in nutrient losses. *Cyanobacteria* can produce microcystin, which is a strong liver tumor promoter, and feeding silage containing *Cyanobacteria* is detrimental to livestock. These four major epiphytic genera were inhibited after 3 days of ensiling, probably due to a rapid decrease in pH, which inhibited their growth, which is beneficial for silage. It is noteworthy that the relative abundance of *Escherichia* in the CK group increased at 30 days of ensiling, accompanied by an increase in pH value and loss of DM, which ultimately caused a decrease in silage quality in the CK group. It is well known that lactic acid bacteria play a crucial role in the silage fermentation process, and different species and characteristics can affect the fermentation quality of the feed. In the CK and CE groups, the predominant genera at silage initiation were *Weissella* and *Pediococcus*. *Weissella*, an early settler in silage, is a heterotypic fermenting lactic acid bacterium, which can produce AA, LA, and CO_2_ using WSC [41]. *Pediococcus* can rapidly grow at high pH and produce lactic acid, inhibiting spoilage microorganisms and improving fermentation quality [42]. After 15 days of ensiling, the species of lactic acid bacteria shifted towards the genus *Lactiplantibacillus*, which is more tolerant to low pH. In the LAB group, the dominant genus at silage initiation was *Lactiplantibacillus* and remained dominant throughout the silage. *Lactiplantibacillus* is a homofermentative lactic acid bacteria with strong acid-producing ability and often becomes the dominant genus in the later stage of various high-quality silages. Compared to the control silage, although the addition of cellulase and LAB had little effect on the composition of the dominant genera of the bacterial community, there was some degree of difference in the relative abundance of the dominant genera. In particular, at 60 days of ensiling, the CE and LAB groups effectively reduced the abundance of *Pediococcus* and increased the abundance of *Lactiplantibacillus* compared to the CK group. This should be caused by the lack of acidic conditions in the CK group.

The difference of bacterial community between additive treatment group and control group during silage was further discussed by LEfSe analysis. In our study, *Bacillus* and *Serratia* were enriched in CE group at 3 days of ensiling and *Sphingomonas* was enriched in CE group at 7 days of ensiling, both of these microorganisms have the ability to degrade fiber. The study of Sethi et al. [43] found that *Bacillus* and *Serratia* can produce cellulase and degrade cellulose effectively. The study of Guo et al. [44] found that *Sphingomonas* has biodegradability to polycyclic aromatic hydrocarbons. This was consistent with the fact that the NDF and ADF contents in CE group were significantly lower than those in CK and LAB groups at the early stage of silage. *Lactiplantibacillus* was enriched in LAB group after 3 days of silage, indicating that the added lactic acid bacteria successfully started fermentation in the early stage of LAB group silage. The presence of undesirable microorganisms such as *Eggerthella* and *Flavonifractor* in the middle and late stages of silage in the CK and LAB groups compared to the CE group indicated that the addition of CE was more effective in suppressing undesirable microorganisms during silage.

The study of the correlation between bacterial communities and silage quality contributes to a comprehensive understanding of the bacteria critical for silage fermentation quality. In this study, *Lactiplantibacillus* was highly negatively correlated with pH and positively correlated with LA content. This is because *Lactiplantibacillus* is a producer of LA, which uses WSC to produce lactic acid and rapidly reduces pH. The genera *Weissella* and *Pediococcus* were also LA producers, while they did not show a positive correlation with LA content and a negative correlation with pH, which could be explained by the weak acid resistance of *Weissella* and *Pediococcus*. Proteases produced by *Enterobacteriaceae* were reported to be associated with NH_3_-N synthesis [45], which explained the positive correlation between NH_3_-N concentration and *Escherichia*, *Enterobacter*, and *Klebsiella* (both *Enterobacteriaceae*). It is known that successful silage is acidic, and dry matter, neutral detergent fiber, and acidic detergent fiber will be decomposed in the acidic environment. In this study, *Lactiplantibacillus* gradually replaced other genera as the dominant genus, and lactic acid and acetic acid produced by WSC lowered the pH, which explained the negative correlation between dry matter, neutral detergent fiber, and acidic detergent fiber and *Lactiplantibacillus*.

Predicting the functional changes of bacterial communities is helpful to evaluate the impact of bacterial communities on the changes of silage quality. Therefore, PICRUSt2 analysis method was used to analyze the relative abundance of KEGG pathway in bacterial communities. On the first pathway level, metabolism was the main metabolic pathway, which indicated that the silage process was mediated by microbial metabolic pathway and could be transformed into different metabolites using fermentable substrates. On the second pathway level, the carbohydrate metabolism, amino acid metabolism, nucleotide metabolism, energy metabolism, and the metabolism of cofactors and vitamins were relatively abundant. Bai et al. [24] found that these metabolic pathways were related to the fermentation quality of silage. Therefore, we further analyzed these metabolic pathways from the level of the third pathway.

It is well known that silage is a process of lactic acid fermentation by lactic acid bacteria using soluble carbohydrates. Soluble carbohydrates are converted into pyruvate by Embden-Meyerhof pathway or pentose phosphate pathway, and lactic acid is formed under the action of lactate dehydrogenase. This study found that the glycolysis/gluconeogenesis pathway and pyruvate metabolism pathway accounted for the highest proportion in the carbohydrate metabolism pathway. Therefore, it can be determined that the fermentation of homolactic acid accounted for a high proportion in the process of *Caragana korshinskii* Silage. The promotion of carbohydrate metabolism after silage may be related to the increase of the relative abundance of lactic acid bacteria. There were no differences in glycolysis/gluconeogenesis metabolic pathways between the three groups, whereas pyruvate metabolism was higher in the CE and LAB groups than in the CK group, which is consistent with the fact that the LA concentration were higher in the CE and LAB groups than in the CK group. It was therefore assumed that promoting the pyruvate metabolic pathway may be a potential way to increase the lactic acid concentration in *Caragana korshinskii* silage. Most of the amino acid metabolic pathways were inhibited after silage, which may be related to the rapid decline of pH in the early fermentation period, limiting the degradation of protein by harmful microorganisms. Notably, the lysine biosynthesis pathway was significantly promoted after silage in each treatment group, which may be related to the increase of *Lactiplantibacillus*. Sands et al. [46] found that *Lactobacillus plantarum* can improve the lysine biosynthesis in silage, and plant feed usually lacks lysine. Therefore, the use of *Lactiplantibacillus plantarum* strains with strong lysine production capacity to prepare silage may have a good application prospect. In this study, the energy metabolism including the oxidative phosphorylation, photosynthesis, photosynthesis proteins, and photosynthesis–antenna proteins were reduced after ensiling. The inhibition of oxidative phosphorylation may be due to the fact that silage is an anaerobic fermentation process, and oxygen was depleted in the early stage of silage, thereby inhibiting this pathway. The inhibition of photosynthesis, photosynthetic proteins, and photosynthesis–antenna proteins was due to a large number of *Cyanobacteria* attached to fresh samples, which were able to perform photosynthesis. As the silage progressed, the *Cyanobacteria* were replaced by lactic acid bacteria, thereby inhibiting this pathway. The porphyrin and chlorophyll metabolism changed the most in the metabolism of cofactors and vitamins pathway, the metabolism of porphyrin and chlorophyll in LAB group was significantly higher than that in other groups. Zhang et al. [47] found that LAB inoculants can promote the production of vitamins in silage. Vitamin B12 is the main product of porphyrin metabolism, and lactic acid bacteria have the ability to produce vitamin B12 [48].

## 5. Conclusions

This study evaluated the effects of cellulase and lactic acid bacteria additives on fermentation parameters, chemical composition, and bacterial community of *Caragana korshinskii* silage. The results of the study showed that the addition of cellulase and lactic acid bacteria improved the ensiling performance of *Caragana korshinskii* by regulating the bacterial community. The addition of cellulase and lactic acid bacteria increased the abundance of *Lactiplantibacillus*, lowered pH, and rapidly inhibited undesirable microorganisms such as *Escherichia*. Compared to lactic acid bacteria additives, the addition of cellulase increased the content of water-soluble carbohydrates in silage and raised the concentration of lactic acid and acetic acid, and therefore cellulase is more suitable for silage *Caragana korshinskii* than lactic acid bacteria. This study reveals the potential of *Caragana korshinskii* as a silage feedstock and provides useful knowledge for the exploitation of the *Caragana korshinskii* resource. However, this study only combined the analysis of silage microorganisms and fermentation quality, which failed to reveal the fermentation mechanism of *Caragana korshinskii* silage in greater depth, and the metabolic process of *Caragana korshinskii* silage needs further study.

## Figures and Tables

**Figure 1 microorganisms-11-00337-f001:**
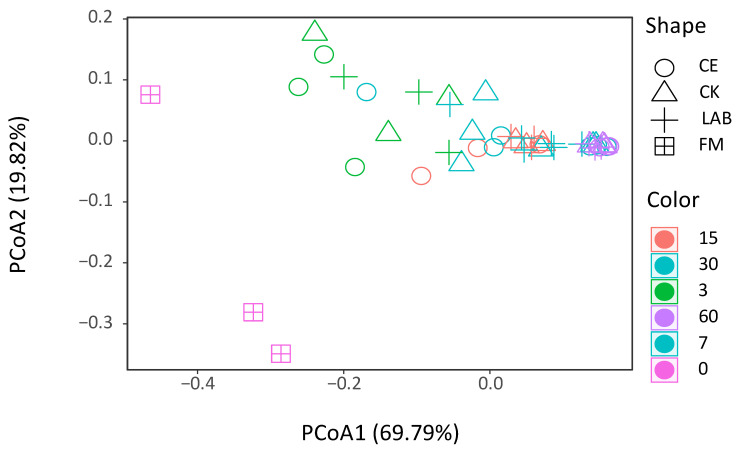
The community dissimilarities in different additives treatments and fermentation time, calculated via weighted UniFrac distances, with coordinates calculated using principal coordinates analysis (PCoA). FM, fresh material; CK, no additive control; CE, cellulase; LAB, lactic acid bacteria.

**Figure 2 microorganisms-11-00337-f002:**
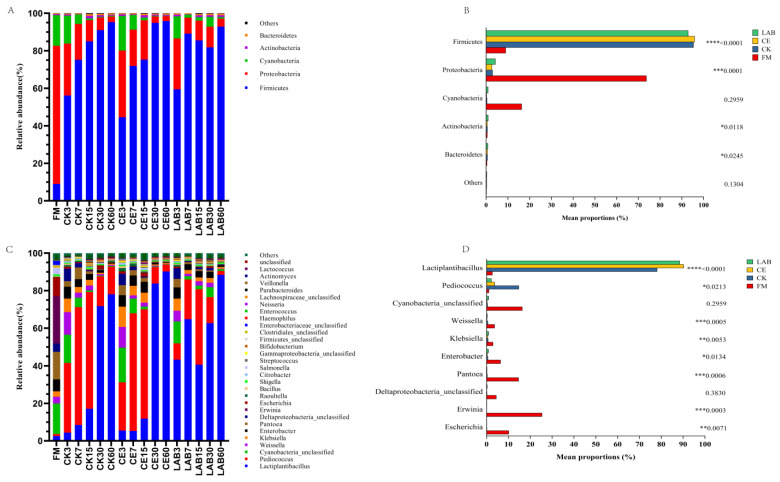
The bacterial community of FM and silage samples. (**A**) The bacterial community was shown at the phylum level. (**B**) One-way analysis of variance bar plots of the phylum level (10 most abundant phylum) among *Caragana korshinskii* silage groups after 60 days of ensiling. (**C**) The bacterial community was shown at the genus level. (**D**) One-way analysis of variance bar plots of the genus level (10 most abundant genera) among *Caragana korshinskii* silage groups after 60 days of ensiling. *, *p* < 0.05; **, *p* < 0.01; ***, *p* < 0.001; ****, *p* < 0.0001; FM, fresh material; CK, no additive control; CE, cellulase; LAB, lactic acid bacteria.

**Figure 3 microorganisms-11-00337-f003:**
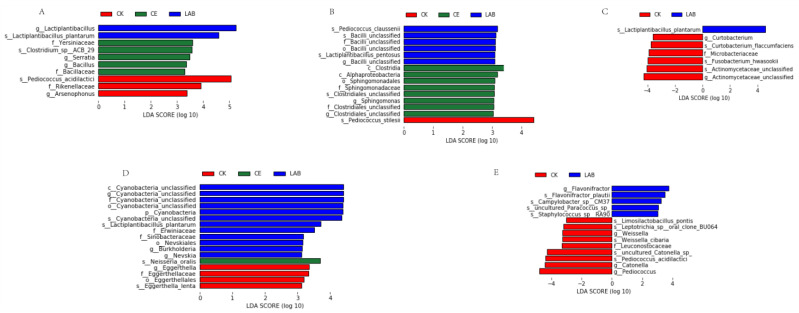
Identification of the communities that have significant differences among the three groups. CK, no additive control; CE, cellulase; LAB, lactic acid bacteria. Arabic number indicates days of ensiling. (**A**) after 3 d of ensiling; (**B**) after 7 d of ensiling; (**C**) after 15 d of ensiling; (**D**) after 30 d of ensiling; (**E**) after 60 d of ensiling.

**Figure 4 microorganisms-11-00337-f004:**
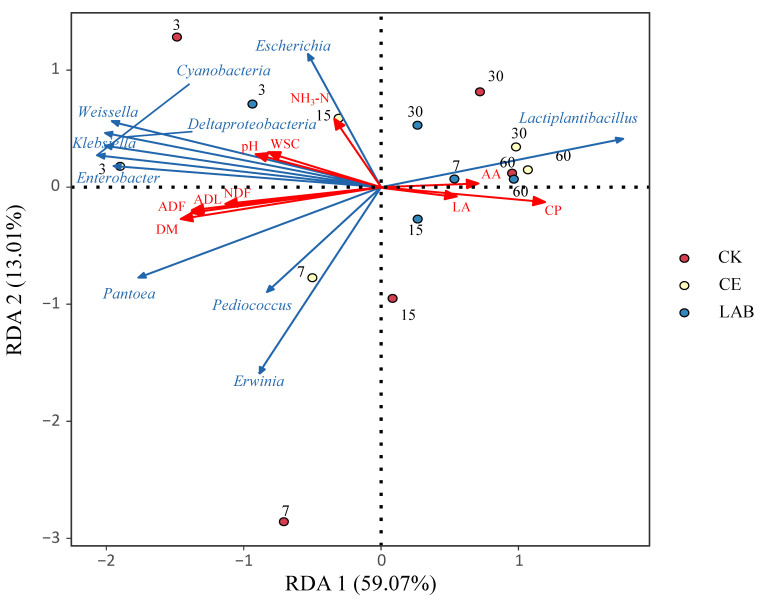
Redundancy analysis (RDA) ordination diagram related to the bacterial with explanatory (e.g., pH; NH_3_-N, LA, AA, DM, NDF, ADF, ADL, CP, WSC) variables represented as arrows by 16S rRNA sequencing. pH, potential of hydrogen; LA, lactic acid; AA, acetic acid; NH_3_-N, ammonia nitrogen; DM, dry matter; NDF, neutral detergent fiber; ADF, acid detergent fiber; ADL, acid detergent lignin; CP, crude protein; WSC, water-soluble carbohydrates.

**Figure 5 microorganisms-11-00337-f005:**
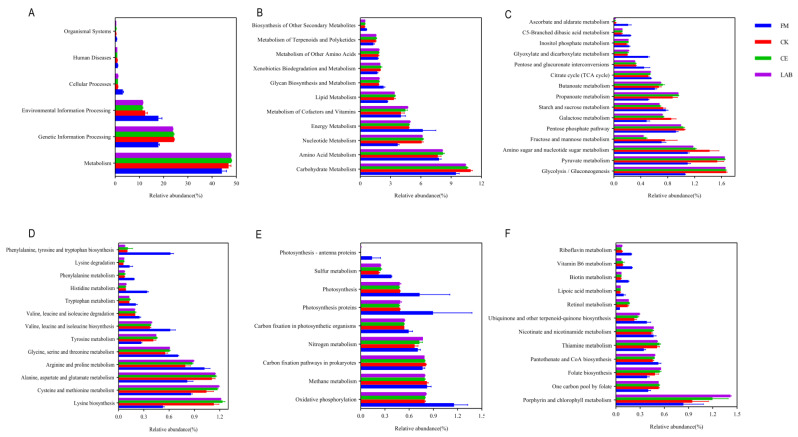
Bar graphs showing 16S rRNA gene-predicted functional profiles for fresh material and after 60 days of ensiling on the first (**A**), second (**B**) pathway levels, carbohydrate metabolism (**C**), amino acid metabolism (**D**), energy metabolism (**E**), and metabolism of cofactors and vitamins (**F**) of the third pathway level. FM, fresh material; CK, no additive control; CE, cellulase; LAB, lactic acid bacteria.

**Table 1 microorganisms-11-00337-t001:** Chemical composition and microbial counts of fresh *Caragana korshinskii*.

Items	Sample
DM (g/kg FM)	536.2 ± 7.31
CP (g/kg DM)	102.53 ± 2.75
WSC (g/kg DM)	27.69 ± 0.69
NDF (g/kg DM)	706.92 ± 2.03
ADF (g/kg DM)	594.99 ± 2.11
ADL (g/kg DM)	155.71 ± 2.22
Lactic acid bacteria (Log_10_ cfu/g FM)	6.11 ± 1.06
Aerobic bacteria (Log_10_ cfu/g FM)	8.15 ± 0.80
Coliform bacteria (Log_10_ cfu/g FM)	7.35 ± 2.13
Yeasts (Log_10_ cfu/g FM)	8.61 ± 0.33
Molds (Log_10_ cfu/g FM)	2.82 ± 0.28

FM, fresh material; DM, dry matter; CP, crude protein; WSC, water soluble carbohydrates; NDF, neutral detergent fiber; ADF, acid detergent fiber; ADL, acid detergent lignin.

**Table 2 microorganisms-11-00337-t002:** Effects of additives and ensiling days on fermentation parameters of *Caragana korshinskii* silage.

Items	Treatments	Ensiling Days					SEM	*p* Value		
		3	7	15	30	60		T	D	T × D
pH value	CK	4.24 ^Aa^	4.02 ^Ab^	3.98 ^Ab^	4.13 ^Aab^	4.12 ^Aab^	0.02	<0.0001	0.048	0.4532
	CE	4.04 ^Ba^	3.89 ^Cb^	3.89 ^Bb^	3.81 ^Bc^	3.74 ^Bd^				
	LAB	4.02 ^Ba^	3.97 ^Bab^	3.95 ^ABbc^	3.9 ^ABc^	3.8 ^Bd^				
LA (g/kg DM)	CK	19.07 ^Aab^	23.87 ^Aab^	24.57 ^Aa^	23.44 ^Aab^	15.26 ^Bb^	0.94	0.0025	0.724	0.0132
	CE	23.81 ^Ab^	24.73 ^Ab^	25.13 ^Ab^	27.92 ^Ab^	38.35 ^Aa^				
	LAB	26.40 ^Aa^	28.06 ^Aa^	26.13 ^Aa^	24.36 ^Aa^	25.19 ^Ba^				
AA (g/kg DM)	CK	7.27 ^Ab^	8.24 ^Ab^	6.61 ^Ab^	12.90 ^Aa^	9.03 ^Bab^	0.63	<0.0001	0.0003	<0.0001
	CE	7.57 ^Ab^	4.94 ^Ab^	5.08 ^Ab^	9.49 ^Ab^	18.37 ^Aa^				
	LAB	3.29 ^Aa^	6.07 ^Aa^	6.12 ^Aa^	3.52 ^Ba^	4.83 ^Ba^				
NH_3_-N (g/kg DM)	CK	0.15 ^Bb^	0.10 ^Ab^	0.09 ^Ab^	0.34 ^Aa^	0.17 ^Ab^	0.03	<0.0001	<0.0001	<0.0001
	CE	0.05 ^Ba^	ND	ND	ND	ND				
	LAB	0.62 ^Aa^	0.11 ^Ac^	ND	0.28 ^Ab^	ND				

DM, dry matter; LA, lactic acid; AA, acetic acid; NH_3_-N, ammonia nitrogen; CK, no additive control; CE, cellulase; LAB, lactic acid bacteria. Different capital letters indicate significant differences among different treatments under the same ensiling days (*p* < 0.05); different lowercase letters indicate significant differences among different ensiling days under the same treatment (*p* < 0.05); same letter indicate not significant (*p* > 0.05); ND, no detected; SEM, standard error of the mean; T, treatments; D, ensiling days; T × D, interaction between treatments and ensiling days.

**Table 3 microorganisms-11-00337-t003:** Effects of additives and ensiling days on chemical composition of *Caragana korshinskii* silage.

Items	Treatments	Ensiling Days					SEM	*p* Value		
		3	7	15	30	60		T	D	T × D
DM (g/kg FM)	CK	395.42 ^Aa^	391.87 ^Ba^	383.42 ^Ab^	369.63 ^Bc^	359.61 ^Bd^	1.73	<0.0001	<0.0001	0.0067
	CE	397.63 ^Aa^	393.78 ^Aba^	385.35 ^Ab^	377.73 ^Abc^	371.69 ^Ac^				
	LAB	398.43 ^Aa^	395.39 ^Aa^	388.76 ^Ab^	380.90 ^Ac^	375.77 ^Ad^				
NDF (g/kg DM)	CK	688.28 ^Aa^	677.67 ^Ab^	670.33 ^Ac^	662.62 ^Ad^	658.47 ^Ad^	2.31	<0.0001	<0.0001	0.0016
	CE	677.67 ^Ba^	660.76 ^Cb^	648.02 ^Cc^	639.27 ^Bd^	635.55 ^Ce^				
	LAB	685.18 ^Aa^	672.22 ^Bb^	666.30 ^Bc^	657.73 ^Ad^	651.08 ^Be^				
ADF (g/kg DM)	CK	582.67 ^Aa^	573.10 ^Ab^	565.65 ^Ac^	556.95 ^Ad^	545.80 ^Ae^	2.62	<0.0001	<0.0001	0.0425
	CE	571.61 ^Ba^	556.17 ^Bb^	544.33 ^Cc^	535.94 ^Cd^	518.88 ^Be^				
	LAB	580.42 ^Aa^	569.53 ^Ab^	561.29 ^Bb^	551.06 ^Bc^	540.74 ^Ad^				
ADL (g/kg DM)	CK	149.05 ^Aa^	145.65 ^Ab^	143.67 ^Abc^	141.79 ^Ac^	139.08 ^Ad^	0.78	<0.0001	<0.0001	0.9621
	CE	142.31 ^Ba^	138.19 ^Bb^	135.69 ^Cc^	133.28 ^Bd^	130.90 ^Be^				
	LAB	148.42 ^Aa^	144.80 ^Aab^	141.16 ^Bbc^	139.77 ^Abc^	138.24 ^Ac^				
CP (g/kg DM)	CK	101.24 ^Bd^	102.60 ^Abc^	103.51 ^Bab^	103.28 ^Cbc^	104.19 ^Ba^	0.27	<0.0001	<0.0001	<0.0001
	CE	102.55 ^Ae^	103.83 ^Ad^	104.28 ^Ac^	105.46 ^Ab^	106.35 ^Aa^				
	LAB	99.63 ^Cd^	101.03 ^Bc^	103.59 ^Bb^	104.29 ^Bb^	105.82 ^Aa^				
WSC (g/kg DM)	CK	23.78 ^Aa^	20.07 ^Bb^	18.66 ^Cbc^	16.14 ^Ccd^	15.70 ^Bd^	0.63	<0.0001	0.0006	0.3776
	CE	28.1 ^Aa^	27.12 ^Aa^	26.53 ^Aa^	26.24 ^Aa^	25.99 ^Aa^				
	LAB	23.95 ^Aa^	21.91 ^Ba^	21.43 ^Ba^	21.17 ^Ba^	20.06 ^Ba^				

FM, fresh material; DM, dry matter; NDF, neutral detergent fiber; ADF, acid detergent fiber; ADL, acid detergent lignin; CP, crude protein; WSC, water-soluble carbohydrates; CK, no additive control; CE, cellulase; LAB, lactic acid bacteria. Different capital letters indicate significant differences among different treatments under the same ensiling days (*p* < 0.05); different lowercase letters indicate significant differences among different ensiling days under the same treatment (*p* < 0.05); same letter indicate not significant (*p* > 0.05). SEM, standard error of the mean; T, treatments; D, ensiling days; T × D, interaction between treatments and ensiling days.

**Table 4 microorganisms-11-00337-t004:** Effects of additives and ensiling days on bacterial alpha diversity of *Caragana korshinskii* silage.

Items	Treatment	Ensiling Days					SEM	*p* Value		
		3	7	15	30	60		T	D	T × D
OTUs	CK	250.67 ^Abc^	222 ^Abc^	318.33 ^Aa^	276.67 ^Aab^	215.33 ^Ac^	6.35	0.513	<0.0001	0.1206
	CE	268.33 ^Aab^	241 ^Abc^	310.33 ^ABa^	222 ^Abc^	188.67 ^Ac^				
	LAB	268 ^Aa^	214.67 ^Ab^	284.67 ^Ba^	265.33 ^Aa^	214.33 ^Ab^				
Chao1	CK	251.21 ^Abc^	222.07 ^Ac^	318.68 ^Aa^	277.17 ^Aab^	215.85 ^Ac^	6.33	0.5146	<0.0001	0.119
	CE	268.48 ^Aab^	241.56 ^Abc^	310.72 ^ABa^	222.61 ^Abc^	189.32 ^Ac^				
	LAB	268.47 ^Aa^	214.77 ^Ab^	284.67 ^Ba^	265.35 ^Aa^	215.14 ^Ab^				
Simpson	CK	0.88 ^ABa^	0.83 ^Aab^	0.86 ^Aab^	0.72 ^Ab^	0.76 ^Aab^	0.03	0.0032	<0.0001	0.0044
	CE	0.91 ^Aa^	0.75 ^Aa^	0.86 ^Aa^	0.51 ^Ab^	0.52 ^Bb^				
	LAB	0.86 ^Ba^	0.74 ^Aab^	0.88 ^Aa^	0.63 ^Ab^	0.29 ^Cc^				
Shannon	CK	4.45 ^Aa^	3.98 ^Aab^	4.29 ^Aab^	3.72 ^Aab^	3.41 ^Ab^	0.16	0.0166	<0.0001	0.0067
	CE	4.98 ^Aa^	3.76 ^Ab^	4.47 ^Aab^	2.27 ^Ac^	2.21 ^Bc^				
	LAB	4.45 ^Aab^	3.49 ^Abc^	4.51 ^Aa^	3.15 ^Ac^	1.53 ^Bd^				
Coverage	CK	0.99	0.99	0.99	0.99	0.99	0	NS	NS	NS
	CE	0.99	0.99	0.99	0.99	0.99				
	LAB	0.99	0.99	0.99	0.99	0.99				
Pielou-e	CK	0.56 ^Aa^	0.51 ^ABa^	0.52 ^ABa^	0.46 ^ABa^	0.44 ^Ba^	0.02	0.0097	<0.0001	0.0046
	CE	0.62 ^Aa^	0.47 ^Ba^	0.54 ^ABa^	0.29 ^Ca^	0.29 ^Cb^				
	LAB	0.55 ^Aa^	0.45 ^ABa^	0.55 ^Aa^	0.39 ^Ba^	0.2 ^Cb^	☐	☐	☐	☐

OTUs, operational taxonomic units; CK, no additive control; CE, cellulase; LAB, lactic acid bacteria. Different capital letters indicate significant differences among different treatments under the same ensiling days (*p* < 0.05); different lowercase letters indicate significant differences among different ensiling days under the same treatment (*p* < 0.05); no or same letter indicate not significant (*p* > 0.05). SEM, standard error of the mean; T, treatments; D, ensiling days; T × D, interaction between treatments and ensiling days.

## Data Availability

The raw sequence data were uploaded to the NCBI archive of sequence reads under study record number PRJNA888202.

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
