# Peer review of "Effects of Cellulase and Lactic Acid Bacteria on Ensiling Performance and Bacterial Community of Caragana korshinskii Silage"

_microorganisms, 2023, doi:10.3390/microorganisms11020337_

Round 1

Reviewer 1 Report

Comments to the Author

MANUSCRIPT DETAILS

Ms. Ref. No. microorganisms-2141900

Title: Effects of Cellulase and Lactic Acid Bacteria on Ensiling Performance and Bacterial Community of Caragana korshinskii Silage

Article Type: Research Article

Journal: microorganisms

GENERAL COMMENTS

The interest in this manuscript is significant enough to merit publication.

My recommendation on submitted manuscript to the Metabolites is to be accepted after minor revisions.

The comments and questions provided below may help the authors to put the manuscript into better appropriate form for publication.

SPECIFIC COMMENTS

·       Line 166 Authors should add SD of result in table 1   

·       Line 183 Authors should add letter of statistical analysis as superscript letter

·       Figure 3 is not clear 

·       Authors should Relationships between Fermentation Parameters and Bacterial Community in detail

Therefore, I think this paper is acceptable for publication with major revision.

Author Response

Response to Reviewer 1 Comments

Dear Reviewer:

Thanks a lot for your review of our manuscript. We will be happy to edit the text further, based on helpful comments from reviewers. We appreciate the editor and reviewers very much for their positive and constructive comments and suggestions. The comments and suggestions are not only helpful for us to revise and improve our manuscript, but also benefit for our further research. We hope that our paper much better quality than before. Now we are resubmitting the revised version after incorporating the comments of the reviewers and making the improvements.

Point 1: Line 166 Authors should add SD of result in table 1.

Response 1: Yes, we have revised in the manuscript.

Items

Sample

DM (g/kg FM)

536.2 ± 7.31

CP (g/kg DM)

102.53 ± 2.75

WSC (g/kg DM)

27.69 ± 0.69

NDF (g/kg DM)

706.92 ± 2.03

ADF (g/kg DM)

594.99 ± 2.11

ADL (g/kg DM)

155.71 ± 2.22

Lactic acid bacteria (Log10 cfu/g FM)

6.11 ± 1.06

Aerobic bacteria (Log10 cfu/g FM)

8.15 ± 0.80

Coliform bacteria (Log10 cfu/g FM)

7.35 ± 2.13

Yeasts (Log10 cfu/g FM)

8.61 ± 0.33

Molds (Log10 cfu/g FM)

2.82 ± 0.28

Point 2: Line 183 Authors should add letter of statistical analysis as superscript letter.

Response 2: Thank you for the suggestion. We have changed all the letters in the table for statistical analysis to superscript letters.

Point 3: Figure 3 is not clear.

Response 3: Thank you for the suggestion. We have redrawn Figure 3. Please see attachment.

Point 4: Authors should Relationships between Fermentation Parameters and Bacterial Community in detail.

Response 4: Thank you for the suggestion. This suggestion is beneficial for our research. We have clearly revised the discussion section as follows :

「The study of the correlation between bacterial communities and silage quality contributes to a comprehensive understanding of the bacteria critical for silage fermentation quality. In this study, Lactiplantibacillus was highly negatively correlated with pH and positively correlated with LA content. This is because Lactiplantibacillus is a producer of LA, which uses WSC to produce lactic acid and rapidly reduces pH. The genera Weissella and Pediococcus were also LA producers, while did not show a positive correlation with LA content and a negative correlation with pH, which could be explained by the weak acid resistance of Weissella and Pediococcus. Proteases produced by Enterobacteriaceae were reported to be associated with NH3-N synthesis [46], which explained the positive correlation between NH3-N concentration and Escherichia, Enterobacter and Klebsiella (both Enterobacteriaceae). It is known that successful silage is acidic, and dry matter, neutral detergent fiber, and acidic detergent fiber will be decomposed in the acidic environment. In this study, Lactiplantibacillus gradually replaced other genera as the dominant genus, and lactic acid and acetic acid produced by soluble sugar lowered the pH, which explained the negative correlation between dry matter, neutral detergent fiber, and acidic detergent fiber and Lactiplantibacillus.」

We would like to thank you again for taking the time to review our manuscript. If we misunderstood the comments or there are still problems with the manuscript, please let us know and we will make further explains or changes. We hope the revised manuscript could be acceptable for you.

Best wishes!

Baochao Bai

E-Mail: bbc66688@163.com

Reviewer 2 Report

The aims of the study are clearly defined both in the abstract and at the end of the introduction.

Please define FM the first time it appears in the text, in line 95.

Please use the new taxonomy of the genus Lactobacillus proposed by Zheng et al. (2020).

Zheng J, Wittouck S, Salvetti E, Franz CMAP, Harris HMB, Mattarelli P, O'Toole PW, Pot B, Vandamme P, Walter J, Watanabe K, Wuyts S, Felis GE, Gänzle MG, Lebeer S. A taxonomic note on the genus Lactobacillus: Description of 23 novel genera, emended description of the genus Lactobacillus Beijerinck 1901, and union of Lactobacillaceae and Leuconostocaceae. Int J Syst Evol Microbiol. 2020 Apr;70(4):2782-2858. doi: 10.1099/ijsem.0.004107

In the caption of Figure 4, please correct "Rredundancy analysis" to "Redundancy analysis".

In the "Conclusions" section, please better explain the claim in the last sentence "In conclusion, both cellulase and lactic acid bacteria could improve the quality of Caragana korshinskii silage, and the addition of cellulase expressed better results than lactic acid bacteria.", coming back to the objectives of your study: "...evaluate the effects of cellulase (CE) and lactic acid bacteria (LAB) on Caragana korshinskii silage by analyzing the fermentation parameters, chemical composition and bacterial community."

Author Response

Response to Reviewer 2 Comments

Dear Reviewer:

Thanks a lot for your review of our manuscript. We will be happy to edit the text further, based on helpful comments from reviewers. We appreciate the editor and reviewers very much for their positive and constructive comments and suggestions. The comments and suggestions are not only helpful for us to revise and improve our manuscript, but also benefit for our further research. We hope that our paper much better quality than before. Now we are resubmitting the revised version after incorporating the comments of the reviewers and making the improvements.

Point 1: Please define FM the first time it appears in the text, in line 95.

Response 1: Yes, we have revised in the manuscript.

「(2) cellulase (CE; Yidu Biological Technology Co., Ltd., Hohhot, Inner Mongolia, China; mainly including exo-β-glucanase, endo-β-glucanase, β-glucosidase and xylanase.; CE activity as 5×105 U/g; addition amount: 1×105 U/kg fresh material (FM)).」

Point 2: Please use the new taxonomy of the genus Lactobacillus proposed by Zheng et al. (2020).

Response 2: Thank you for your suggestion. This suggestion is useful for our study. The genus Lactobacillus determined in this study belongs to the genus Lactiplantibacillus in the new taxonomy, and we have changed the genus of Lactobacillus in the text and redrawn the diagram.

Point 3: In the caption of Figure 4, please correct "Rredundancy analysis" to "Redundancy analysis".

Response 3: Thanks for your careful check. We have corrected the word.

「Redundancy analysis (RDA) ordination diagram related to the bacterial with explanatory (e.g., pH; NH3-N, LA, AA, DM, NDF, ADF, ADL, CP, WSC) variables represented as arrows by 16S rRNA sequencing. pH, potential of hydrogen.」

Point 4: In the "Conclusions" section, please better explain the claim in the last sentence "In conclusion, both cellulase and lactic acid bacteria could improve the quality of Caragana korshinskii silage, and the addition of cellulase expressed better results than lactic acid bacteria.", coming back to the objectives of your study: "...evaluate the effects of cellulase (CE) and lactic acid bacteria (LAB) on Caragana korshinskii silage by analyzing the fermentation parameters, chemical composition and bacterial community."

Response 4: Thank you for the suggestion. The comments and suggestions are beneficial for our research. We have revised clearly as:

「This study evaluated the effects of cellulase and lactic acid bacteria additives on fermentation parameters, chemical composition and bacterial community of Caragana korshinskii silage. The results of the study showed that the addition of cellulase and lactic acid bacteria improved the ensiling performance of Caragana korshinskii by regulating the bacterial community. The addition of cellulase and lactic acid bacteria increased the abundance of Lactiplantibacillus, lowered pH and rapidly inhibited undesirable microorganisms such as Escherichia. Compared to lactic acid bacteria additives, the addition of cellulase increased the content of water-soluble carbohydrates in silage and raised the concentration of lactic acid and acetic acid, and therefore cellulase is more suitable for silage Caragana korshinskii than lactic acid bacteria. This study reveals the potential of Caragana korshinskii as a silage feedstock, and provides useful knowledge for the exploitation of the Caragana korshinskii resource. However, this study only combined the analysis of silage microorganisms and fermentation quality, which failed to reveal the fermentation mechanism of Caragana korshinskii silage in greater depth, and the metabolic process of Caragana korshinskii silage needs further study.」

We would like to thank you again for taking the time to review our manuscript. If we misunderstood the comments or there are still problems with the manuscript, please let us know and we will make further explains or changes. We hope the revised manuscript could be acceptable for you.

Best wishes!

Baochao Bai

E-Mail: bbc66688@163.com

Reviewer 3 Report

Dear authors,

this paper is interesting and scientifically sounds. Otherwise, this plant (Caragana korshinskii) as far as I know is scarcely known in Europe for example, thus it would be nice if the authors can introduce it in introduction section.

Keywords: for higher indexing, please avoid to use words already present in the title.

The use of acronyms should be limited in the text, although it can be used more intensely on the figures and tables, with abbreviations reported on the captions. The acronyms in the text should be related to the studied cases and to a limited number of variables.

Line 111: MRS what’s for? Please report the corrrect names of media and the suppliers details.

potato glucose agar medium,Luria-Bertani medium, nutrient agar medium….Please report the exact name and supplier. If not a n original recipe was used, then report the single ingredients concentrations

Line 116: lg cfu/g ??? you do meant Log10 CFU/g

Line 117. Avoid the use of the verb tense “shall”

Line 125: CTAB..please report the acronym and a reference,

Line 131, 132, 134, and other else: Beckman Coulter, Invitrogen (Thermo Fisher, Waltham, MA, USA)…please report correct supplier’s details the first time it appears

2.4. QIME2, PICRUST, VSearch and others need references. Please report them all correctly.

Tables 2, 3,4  Letters for post hoc must goes in upperscript.

Tables 2, 3,4  Report in the footnotes the explanations of all the acronyms used

Figure 1: The caption is uncomplete, what’s 15, 30, 3, 60, …stands for? Please report it.

Line 240: Cyanobacteria_unclassified….unclassified does not go in italics

Line 243: obvious…please remove and change such not scientific term

Conclusion is poor and should be reinforced with critical evaluations of the work presented, including also pros and weakness of the present work and future presepctives.

Author Response

Response to Reviewer 3 Comments

Dear Reviewer:

Thanks a lot for your review of our manuscript. We will be happy to edit the text further, based on helpful comments from reviewers. We appreciate the editor and reviewers very much for their positive and constructive comments and suggestions. The comments and suggestions are not only helpful for us to revise and improve our manuscript, but also benefit for our further research. We hope that our paper much better quality than before. Now we are resubmitting the revised version after incorporating the comments of the reviewers and making the improvements.

Point 1: This plant (Caragana korshinskii) as far as I know is scarcely known in Europe for example, thus it would be nice if the authors can introduce it in introduction section.

Response 1: Thank you for the suggestion. The comments and suggestions are beneficial for our research. We have added a description of this plant in the introduction section:

Caragana korshinskii is a genus of Caragana in the legume family. The genus Caragana is a dominant species in cold-temperate arid scrub, desert and mountain meadows, and is well known in China for its role in sand fixation, soil and water conservation, nectar, fuel and fodder. To date, more than 100 species of Caragana have been found worldwide, mainly in arid and semi-arid regions of Asia and Europe, while at least 60 species of Caragana have been found in China alone. Among these plants, Caragana korshinskii is the most common plant in the arid and semi-arid regions of the Inner Mongolia Plateau [2].」

Point 2: Keywords: for higher indexing, please avoid to use words already present in the title.

Response 2: Thank you for the suggestion. We have revised clearly as:

Keywords: Lactic acid bacteria; Cellulase; Microbial community; Fermentation characteristics; Caragana korshinskii

Point 3: The use of acronyms should be limited in the text, although it can be used more intensely on the figures and tables, with abbreviations reported on the captions. The acronyms in the text should be related to the studied cases and to a limited number of variables.

Response 3: Thank you for the suggestion. We have changed the acronyms in the text to reduce the use of acronyms.

Point 4: Line 111: MRS what’s for? Please report the correct names of media and the suppliers details. potato glucose agar medium, Luria-Bertani medium, nutrient agar medium….Please report the exact name and supplier. If not a n original recipe was used, then report the single ingredients concentrations.

Response 4: Yes, we have revised.

「Lactic acid bacteria were cultured and counted on an MRS medium. Aerobic bacteria were cultured and counted using a nutrient agar medium. Molds and Yeasts were cultured and counted on potato dextrose agar medium. Coliform bacteria were cultured and counted on eosin methylene blue medium. All four culture media were purchased from Guangdong Huankai Microbial Sci.&Tech.CO.,Ltd.」

Point 5: Line 116: lg cfu/g ??? you do meant Log10 CFU/g.

Response 5: Yes, we have revised.

「Colonies were calculated as the viable number of microorganisms and transformed into logarithmic scales based on FM basis (Log10 cfu/g FM).」

Point 6: Line 117. Avoid the use of the verb tense “shall”.

Response 6: Yes, we have revised.

「Each group of samples was tested 3 times in parallel for chemical analysis.」

Point 7: Line 125: CTAB..please report the acronym and a reference.

Response 7: Yes, we have revised.

「Cetyltriethyl Ammnonium bromide (CTAB) method was used to extract the total DNA of microbial groups from various sources [16], and the quality of DNA extraction was detected by agarose gel electrophoresis.」

Point 8: Line 131, 132, 134, and other else: Beckman Coulter, Invitrogen (Thermo Fisher, Waltham, MA, USA)...please report correct suppliers details the first time it appears.

Response 8: Yes, we have revised.

「The PCR product were purified using the AMPure XT beads (Beckman Coulter Genomics, Danvers, MA, United States), quantified using Qubit (Invitrogen, Carlsbad, CA, United States), then mixed according to the sequencing requirements of each sample. PCR products were detected by 2% agarose gel and recovered by AMPure XT beads recovery kit. The PCR products were sequenced on Agilent 2100 biological analyzer (Agilent Technologies, Inc, Santa Clara, CA, United States) and an Illumina platform (Kapa Biosciences, Woburn, MA, United States) and after edulcoration and ration.」

Point 9: 2.4. QIME2, PICRUST, VSearch and others need references. Please report them all correctly.

Response 9: Yes, we have revised.

「The raw reads were carried out quality filtering by the software of fqtrim (v0.94) (http://ccb.jhu.edu/software/fqtrim/) under specific filtering conditions so that gain high quality clean labels. Using Vsearch software (v2.3.4) (https://github.com/torognes/vsearch) to filter chimeric sequences. Alpha diversity and beta diversity were reckoned by random normalization to the same sequence using QIIME2 [17]. Microbial relative abundance was used to represent bacterial classification. Based on 16S rRNA data, PICRUSt was used to predict metabolic genes [18].」

Point 10: Tables 2, 3,4  Letters for post hoc must goes in upperscript.

Response 10: Thank you for the suggestion. We have changed all the letters in the table for statistical analysis to superscript letters.

Point 11: Tables 2, 3,4  Report in the footnotes the explanations of all the acronyms used.

Response 11: Yes, we have revised.

Point 12: Figure 1: The caption is uncomplete, what’s 15, 30, 3, 60,...stands for? Please report it.

Response 12: Yes, we have revised.

Figure 1. The community dissimilarities in different additives treatments and fermentation time, calculated via weighted UniFrac distances, with coordinates calculated using principal coordinates analysis (PCoA). FM, fresh material; CK, no additive control; CE, cellulase; LAB, Lactic acid bacteria.」

Point 13: Line 240: Cyanobacteria_unclassified...unclassified does not go in italics.

Response 13: Yes, we have revised.

「The main genera attached to fresh Caragana korshinskii were Erwinia (25.32%) and Cyanobacteria_unclassified (16.31%), followed by Pantoea (14.59%) and Escherichia (10.10%) (Figure 2C).」

Point 14: Line 243: obvious...please remove and change such not scientific term.

Response 14: Yes, we have revised.

「The bacterial community composition was similar in CE and CK groups, and the relative abundances of Firmicutes increased with the prolongation of silage time.」

Point 15: Conclusion is poor and should be reinforced with critical evaluations of the work presented, including also pros and weakness of the present work and future presepctives.

Response 15: Yes, we have revised.

「This study evaluated the effects of cellulase and lactic acid bacteria additives on fermentation parameters, chemical composition and bacterial community of Caragana korshinskii silage. The results of the study showed that the addition of cellulase and lactic acid bacteria improved the ensiling performance of Caragana korshinskii by regulating the bacterial community. The addition of cellulase and lactic acid bacteria increased the abundance of Lactiplantibacillus, lowered pH and rapidly inhibited undesirable microorganisms such as Escherichia. Compared to lactic acid bacteria additives, the addition of cellulase increased the content of water-soluble carbohydrates in silage and raised the concentration of lactic acid and acetic acid, and therefore cellulase is more suitable for silage Caragana korshinskii than lactic acid bacteria. This study reveals the potential of Caragana korshinskii as a silage feedstock, and provides useful knowledge for the exploitation of the Caragana korshinskii resource. However, this study only combined the analysis of silage microorganisms and fermentation quality, which failed to reveal the fermentation mechanism of Caragana korshinskii silage in greater depth, and the metabolic process of Caragana korshinskii silage needs further study.」

We would like to thank you again for taking the time to review our manuscript. If we misunderstood the comments or there are still problems with the manuscript, please let us know and we will make further explains or changes. We hope the revised manuscript could be acceptable for you.

Best wishes!

Baochao Bai

E-Mail: bbc66688@163.com

Round 2

Reviewer 1 Report

Accept in present form

Reviewer 3 Report

The revision done are fine for me, the paper can be accepted now. I thank the authors.